# Determinants of care-seeking behavior for sexually transmitted infections among sexually active men in East Africa: A multilevel mixed effect analysis

**Beminate Lemma Seifu**[1]*, **Bezawit Melak Fente**[2], **Mamaru Melkam**[3], **Zufan Alamrie Asmare**[4], **Angwach Abrham Asnake**[5], **Meklit Melaku Bezie**[6], **Hiwot Altaye Asebe**[1], **Yohannes Mekuria Negussie**[7]

1 Department of Public Health, College of Medicine and Health Sciences, Samara University, Samara, Ethiopia, 2 Department of General Midwifery, School of Midwifery, College of Medicine & Health Sciences, University of Gondar, Gondar, Ethiopia, 3 Department of Psychiatry, College of Medicine and Health Science, University of Gondar, Gondar, Ethiopia, 4 Department of Ophthalmology, School of Medicine and Health Science, Debre Tabor University, Debre Tabor, Ethiopia, 5 Department of Epidemiology and Biostatistics, School of Public Health, College of Medicine and Health Sciences, Wolaita Sodo University, Wolaita Sodo, Ethiopia, 6 Department of Public Health Officer, Institute of Public Health, College of Medicine and Health Sciences, University of Gondar, Gondar, Ethiopia, 7 Department of Medicine, Adama General Hospital and Medical College, Adama, Ethiopia

* beminetlemma1915@gmail.com

## Abstract

### Background

Addressing the global challenge of sexually transmitted infections (STIs) is crucial and demands immediate attention. Raising awareness, improving healthcare facilities, and implementing preventive measures are necessary to reduce the spread and mitigate their adverse effects. The treatment seeking behavior of individuals in relation to STIs is an important factor in STI prevention and control. Thus, this study aimed to identify factors associated with STI-related care-seeking behavior among sexually active men in East Africa.

### Methods

A weighted sample of 3,302 sexually active men from recent Demographic and Health Surveys (DHSs) in East African countries were included for analysis. To accommodate the inherent clustering in DHS data and the binary nature of the dependent variable, we applied a multi-level mixed-effect logistic regression model. The deviance value was used to select the best-fitted model. The strength of the association was estimated using an adjusted odds ratio, along with a 95% confidence interval, and statistical significance was determined at a p-value < 0.05.

### Result

The pooled prevalence of STI-related care-seeking behavior among sexually active men in East Africa was 71% (95%CI: 69.76, 72.75). In the multivariable multilevel model,

**Data Availability Statement:** The manuscript contains all result-based data. In addition, the

dataset is publicly available and accessible to the MEASURE DHS Program through https://dhsprogram.com/data/dataset_admin/index.cfm.

**Funding:** The author(s) received no specific funding for this work.

**Competing interests:** The authors have declared that no competing interests exist.

**Abbreviations:** AIDS, Acquired immunodeficiency virus; AOR, Adjusted Prevalence Ratio; CI, Confidence interval; DHS, Demographic and Health Survey; EA, East African; EA, : Enumeration Area; HIV, Human immunodeficiency virus; ICC, Intra-class correlation coefficient; LR, : likelihood Ratio; MOR, Median odds ratio; SR-STIs, self-reported STIs; STIs, Sexually transmitted infections; WHO, World Health Organization.

individuals in the age groups of 25–34 (AOR = 1.58, 95%CI: 1.22, 2.04) and 44 years and above (AOR = 1.44, 95%CI: 1.01, 2.02), those who were married (AOR = 1.62, 95%CI: 1.25, 2.11), had 1 (AOR = 1.88, 95%CI: 1.50, 2.35) and $\geq$2 (AOR = 2.53, 95%CI: 1.89, 3.39) sexual partners excluding their spouse, had ever been tested for HIV (AOR = 1.86, 95%CI: 1.52, 2.28), and had media exposure (AOR = 1.30, 95%CI: 1.04, 1.62) had a positive association with care-seeking behavior for STIs.

## Conclusion

Based on our findings, seven out of ten sexually active men in East Africa exhibit care-seeking behavior for STIs. It is crucial to implement policies and strategies aimed at improving the health-seeking habits of young, unmarried men. Utilizing diverse media platforms to disseminate accurate information and success stories about STI symptoms is pivotal in achieving this goal.

## Introduction

The global prevalence of sexually transmitted infections (STIs) is rising despite the adoption of preventive measures such as comprehensive sexual education, condom distribution programs, regular screening and testing, partner notification and treatment, and public awareness campaigns [1–3]. This ongoing increase highlights a significant and persistent public health challenge that affects myriad individuals worldwide, imposing a substantial burden that resonates globally [4–7]. According to data provided by the World Health Organization (WHO), there are an estimated 374 million annual cases of treatable STIs. Emphasizing the specific burden shouldered by Sub-Saharan Africa (SSA), which comprises 40% of the total global burden of STIs, is essential [1, 8].

In resource-limited nations, inadequate access to equipment, laboratory services, and skilled professionals exacerbates the problem [9]. STIs are generally treatable, but their insidious character, marked by subtle symptoms, presents a hidden threat to both physical and mental well-being [10]. The impact goes beyond the evident, encompassing issues like blindness, cardiovascular diseases, urethral strictures, genital cancers, infertility, and an elevated vulnerability to HIV infection [4, 11]. In low- and middle-income nations, the spectrum of STIs and their associated complications ranks among the primary reasons prompting individuals to seek medical care [12, 13]. Timely identification and intervention in cases of STIs are pivotal in reducing their prevalence and interrupting the transmission cycle [14, 15]. Treatment seeking behavior is an important factor in contributing to the overall effectiveness of STI prevention and control. Comprehending the behavior of seeking treatment for STIs is crucial for controlling their spread since prompt and suitable treatment can shorten infectious periods. Health-seeking behaviors encompass the efforts made by individuals to discern suitable solutions when confronted with illnesses or health-related concerns [16–20].

Despite being the most educated segment of the population, men still face barriers to seeking STI care. Notably, men's reluctance to seek treatment at public health facilities stems from various sources, including societal taboos, personal reservations about sexual health discussions, and privacy concerns. Moreover, the perception of autonomy in sexual matters among men may further deter them from seeking care [17, 21, 22]. This can adversely affect not only the individuals themselves but also their partners, given men typically hold autonomy in sexual matters [23, 24].

Previous studies examining care-seeking behavior for STIs have predominantly centered around women, emphasizing a noticeable gap in the current body of literature regarding the care-seeking practices of men concerning STIs. As early detection and treatment constitute fundamental aspects of STI prevention and control, it is imperative to possess a comprehensive understanding of the factors influencing care-seeking behavior for STIs and the associated potential risks. Therefore, this study aimed to assess the pooled prevalence of STI-related care-seeking behavior and associated factors among sexually active men in East Africa. The findings from this study will be beneficial in understanding men's STI care-seeking practices and the underlying factors influencing them. The insights gained will play a pivotal role in designing innovative policies and strategies tailored to the effective prevention and management of STIs, as well as improving the accessibility and acceptability of STI care services in the East African context.

## Methods

### Study design and setting

This study was conducted utilizing the latest Demographic and Health Survey (DHS) data from East African countries spanning the period from 2011 to 2021. The DHS is a nationally representative survey designed to gather information on key indicators related to population dynamics, nutrition, and health using a community-based cross-sectional study design. A two-stage stratified sampling method was employed to identify participants for the study. In the initial stage, Enumeration Areas (EAs) were randomly chosen based on recent population data, utilizing the housing census as a sampling frame. Subsequently, households were selected in the second stage. For this particular study, focusing on sexually active men, the dataset derived from men's records (MR) file was utilized. Additional information on the DHS methodology is available at https://dhsprogram.com/Methodology/index.cfm.

### Data source and study population

This study utilized recent, nationally representative DHS data from eight countries: Burundi, Congo, Ethiopia, Comoros, Madagascar, Rwanda, Zambia, and Zimbabwe. In East Africa, only 15 countries have a documented history of involvement in the DHS. Notably, countries like Sudan and Eritrea did not have recent DHS data available. Additionally, Uganda, Mozambique, and Malawi lacked recorded information on STIs among sexually active men, while Tanzania and Kenya do not incorporate variables related to care-seeking behavior in their DHS datasets. In this study, a total weighted sample of 3302 sexually active men was considered in the final analysis.

### Study variables

**Dependent variable.**   The DHS survey defined STI as the "Percentage of men aged 15–49 who had an STI in the past 12 months, had an abnormal genital discharge in the past 12 months, had a genital sore or ulcer in the past 12 months, had an STI or symptoms of an STI in the past 12 months.

*The outcome variable is defined as.* Among men aged 15–49 who reported a sexually transmitted infection (STI) or symptoms of an STI in the past 12 months, the percentage who sought advice or treatment. Individuals who have sought medical treatment or advice for sexually transmitted infections (STIs) were categorized as having STI-related care-seeking behavior and designated as "Yes". Conversely, individuals who exhibit symptoms of STIs or have STIs but do not seek medical treatment or advice are classified as not having STI-related care-seeking behavior and marked as "No" [25].

**Independent variables.** In the current study, aligning with the study's objectives and acknowledging the hierarchical nature of the DHS data, two levels of independent variables were considered: individual and community-level variables. We have chosen the potential predictors of the outcome variable based on the reviewed works of literature [12, 14, 16, 17, 20, 22, 26, 27] and their clinical importance.

Age, age at first sex, educational status, marital status, employment status, wealth status, exposure to mass media, number of sexual partners in the last 12 months excluding the spouse, ever heard about STI, ever been tested for HIV, comprehensive HIV and AIDS knowledge, and health insurance coverage were individual-level variables. The community-level variables were place of residence and East African countries.

*Wealth index.* The wealth index measures households' relative wealth using data on assets, housing, water access, and sanitation. A statistical procedure called principal components analysis is used to place households on a continuous scale of relative wealth. The index is used to compare wealth's influence on population, health, and nutrition indicators, with households divided into five wealth quintiles (poorest, poor, middle, rich and richest). The wealth index is presented in DHS reports and survey datasets as a background characteristic https://www. dhsprogram.com/topics/wealth-index/.

*Media exposure.* This study measures media exposure through three variables: how often a person listens to the radio, watches television, or reads newspapers/magazines. If a Man engages in any of these activities at least once a week, he is considered to have media exposure (coded as "Yes"). On the other hand, if he does not engage in any of these activities, he is considered to have no media exposure (coded as "No").

**Data processing and statistical analysis.** STATA version 17 statistical software was employed for data extraction, recoding, and analysis. The data were weighted based on sampling weight, primary sampling unit, and strata to restore its representativeness and account for the sampling design when computing standard errors, ensuring reliable statistical estimates. In the realm of descriptive statistics, we have utilized a weighted sample. This approach has assigned a weight to each observation in our dataset, accounting for its importance or representativeness. However, when generating a forest plot to estimate the pooled prevalence, we did not use a weighted sample. This was because the meta-analysis itself provides a weight for each country, which subsequently generates the pooled estimate. Descriptive statistics were utilized to portray the study population with respect to pertinent characteristics. We conducted a meta-analysis by combining prevalence estimates from different countries to obtain a pooled estimate. A multilevel binary logistic regression model was applied to account for the hierarchical nature of the DHS data. Bivariable multilevel binary logistic regression analysis was performed to identify variables eligible for the multivariable analysis at a p-value < 0.20 (S1 Table).

Following the selection of variables for the multivariable multilevel binary logistic regression analysis, four models comprising the identified variables were constructed. The first model was a null model without independent variables to determine the extent of cluster variation. The second model incorporated individual-level variables, the third involved community-level variables, and the fourth model simultaneously considered the effects of both individual and community-level variables to see which model will explain our data better.

The Likelihood Ratio (LR) test and intra-class Correlation Coefficient (ICC) were computed to assess the variability in STI-related care-seeking behavior among clusters or communities. Deviance was utilized to validate the model's fitness, and the model with the lowest deviance was considered the best-fitted model. In the final model, the Adjusted Odds Ratio (AOR) along with its 95% confidence interval (CI) was employed to estimate the strength of

association between individual and community-level characteristics with self-reported STI. At this level, variables with a p-value < 0.05 were considered statistically significant.

**Ethical consideration.** As the study was a secondary data analysis of publicly accessible survey data from the MEASURE DHS program, this study did not require ethical approval and participant consent. We have granted permission from http://www.dhsprogram.com to download and use the data for this study. In the datasets, there are no names of persons or household addresses recorded.

## Result

### Descriptive characteristics of the study participants

Among the 3,302 sexually active men who had STI in the past 12 months, 29.67% of them were between the ages of 15–24 while 34.51% of them were between the ages of 25–34. The vast majority (68.13%) of them were married and 44.50% of them had a secondary educational status. Regarding age at first sex; 72.95% had sex before the age of 19 and 55.27% ever tested for HIV. More than two-thirds (78.99%) of the study participants had media exposure and 61.34% reside in rural households (Table 1).

### The pooled prevalence of sexually transmitted infections related care seeking behavior and its distribution across independent variables

The pooled prevalence of STI-related health care-seeking behavior among men in East Africa was 71% (95%CI: 69.76, 72.75). The highest prevalence of healthcare-seeking behavior was observed in Congo at 88.43% (95%CI: 85.73, 91.14) and the lowest was in Ethiopia at 44.54% (95%CI: 38.10, 50.98) (Fig 1).

### Statistical analysis and model comparison

Even though, the ICC value was less than 10% the Log-likelihood Ratio (LR) was significant, indicating that a multilevel binary logistic regression model better fits the data than the classical regressions. The Log-likelihood ratio test which was ($X^2$ = 29.11, $p$-value < 0.001) informed us to choose the generalized linear mixed-effect model (GLMM) over the basic model. The models were compared with deviance and the final model with both individual and community-level variables was chosen as the best-fitted model since it had the lowest deviance value (3,442.122) (Table 2).

### Determinants of STI-related health care-seeking behavior among sexually active men in East Africa

In the multivariable multilevel model; age, marital status, number of sexual partners, ever been tested for HIV, media exposure, residing in Congo and Madagascar were associated with statistically significant higher odds of healthcare-seeking behavior for STI. However, residing in Burundi, Ethiopia, and Zimbabwe was associated with lower odds of healthcare-seeking behavior towards STI.

Compared to adolescent men the odds of healthcare-seeking behavior towards STI were 1.58 times (AOR = 1.58, 95%CI: 1.22, 2.04) and 1.44 times (AOR = 1.44, 95%CI: 1.01, 2.02) times higher among men whose age is between 25–34 and >44, respectively. Married men had 1.62 times (AOR = 1.62, 95%CI: 1.25, 2.11) increased odds of seeking medical care for STI symptoms compared to single men. Men who had 1 and ≥2 sexual partners excluding their spouse were 1.88 times (AOR = 1.88, 95%CI: 1.50, 2.35) and 2.53 times (AOR = 2.53, 95%CI: 1.89, 3.39) at higher odds of health care seeking for STI. The odds of healthcare-seeking

**Table 1. Socio-demographic characteristics of the study participants and the prevalence of sexually transmitted infections across independent the variables (n = 3,302).**

| Variable | Weighted frequency (%) | Self-reported sexually transmitted infection | |
|---|---|---|---|
| | | No (%) | Yes (%) |
| **Individual level characteristics** | | | |
| **Age** | | | |
| 15–24 | 980 (29.67) | 403 (41.12) | 576 (58.88) |
| 25–34 | 1,140 (34.51) | 327 (28.69) | 812 (71.31) |
| 35–44 | 743 (22.51) | 245 (32.99) | 498 (67.01) |
| >44 | 439 (13.31) | 184 (41.97) | 255 (58.03) |
| **Marital status** | | | |
| Never married | 1,052 (31.87) | 380 (36.10) | 672 (63.90) |
| Ever married | 2,249 (68.13) | 779 (34.65) | 1,470 (65.35) |
| **Educational status** | | | |
| No formal education | 386 (11.69) | 171 (44.21) | 215 (55.79) |
| Primary | 1,243 (37.65) | 479 (38.58) | 763 (61.42) |
| Secondary | 1,469 (44.50) | 444 (30.24) | 1,025 (69.76) |
| Higher | 203 (6.16) | 64 (31.75) | 139 (68.25) |
| **Wealth index** | | | |
| Poor | 1,132 (34.27) | 405 (35.81) | 726 (64.19) |
| Middle | 728 (22.05) | 252 (34.63) | 476 (65.37) |
| Rich | 1,442 (43.68) | 502 (34.81) | 940 (65.19) |
| **Employment status** | | | |
| Not employed | 444 (13.46) | 160 (36.05) | 284 (63.95) |
| Employed | 2,858 (86.54) | 999 (34.97) | 1,858 (65.03) |
| **Age at first sex** | | | |
| ≤ 19 | 2,409 (72.95) | 777 (32.26) | 1,631 (67.74) |
| ≥ 20 | 893 (27.05) | 382 (42.81) | 511 (57.19) |
| **Number of sex partners excluding spouse, in the last 12 months** | | | |
| 0 | 1,602 (48.52) | 724 (45.18) | 878 (54.82) |
| 1 | 1,021 (30.91) | 276 (27.07) | 744 (72.93) |
| ≥2 | 679 (20.57) | 159 (23.47) | 520 (76.53) |
| **Ever heard about STI** | | | |
| No | 21 (0.62) | 7 (32.77) | 14 (62.23) |
| Yes | 3,281 (99.38) | 1,152 (35.13) | 2,129 (64.87) |
| **Ever heard of AIDS** | | | |
| **No** | 102 (3.09) | 30 (29.32) | 72 (70.68) |
| **Yes** | 3,200 (96.91) | 1,130 (35.30) | 2,070 (64.70) |
| **Ever been tested for HIV** | | | |
| No | 1,477 (44.73) | 575 (38.93) | 902 (61.07) |
| Yes | 1,825 (55.27) | 584 (32.02) | 1,241 (67.98) |
| **Media exposure** | | | |
| No | 694 (21.01) | 283 (40.91) | 410 (59.09) |
| Yes | 2,608 (78.99) | 876 (33.57) | 1,732 (66.43) |
| **Comprehensive knowledge about HIV/AIDS (n = 2,838)** | | | |
| No | 190 (6.71) | 93 (48.80) | 97 (51.20) |
| Yes | 2,648 (93.29) | 956 (36.09) | 1,692 (63.91) |
| **Covered by health insurance** | | | |
| No | 2,850 (86.31) | 1,008 (35.35) | 1,842 (64.65) |

(*Continued*)

**Table 1.** (Continued)

| Variable | Weighted frequency (%) | Self-reported sexually transmitted infection | |
|---|---|---|---|
| | | No (%) | Yes (%) |
| Yes | 452 (13.69) | 152 (33.60) | 300 (66.40) |
| **Community level characteristics** | | | |
| **Residence** | | | |
| Urban | 1,276 (38.66) | 381 (29.88) | 895 (70.12) |
| Rural | 2,026 (61.34) | 778 (38.41) | 1,247 (61.59) |

behavior among men who had ever tested for HIV and had media exposure were 1.86 times (AOR = 1.86, 95%CI: 1.52, 2.28) and 1.30 times (AOR = 1.30, 95%CI: 1.04, 1.62) higher, respectively. Lower odds of STI related care seeking behavior were observed in Burundi (AOR = 0.56, 95%CI: 0.39, 0.80), Ethiopia (AOR = 0.41, 95%CI: 0.28, 0.59), and, Zimbabwe (AOR = 0.28, 95%CI: 0.22, 0.37); while increased odds were observed in Congo (AOR = 3.55,

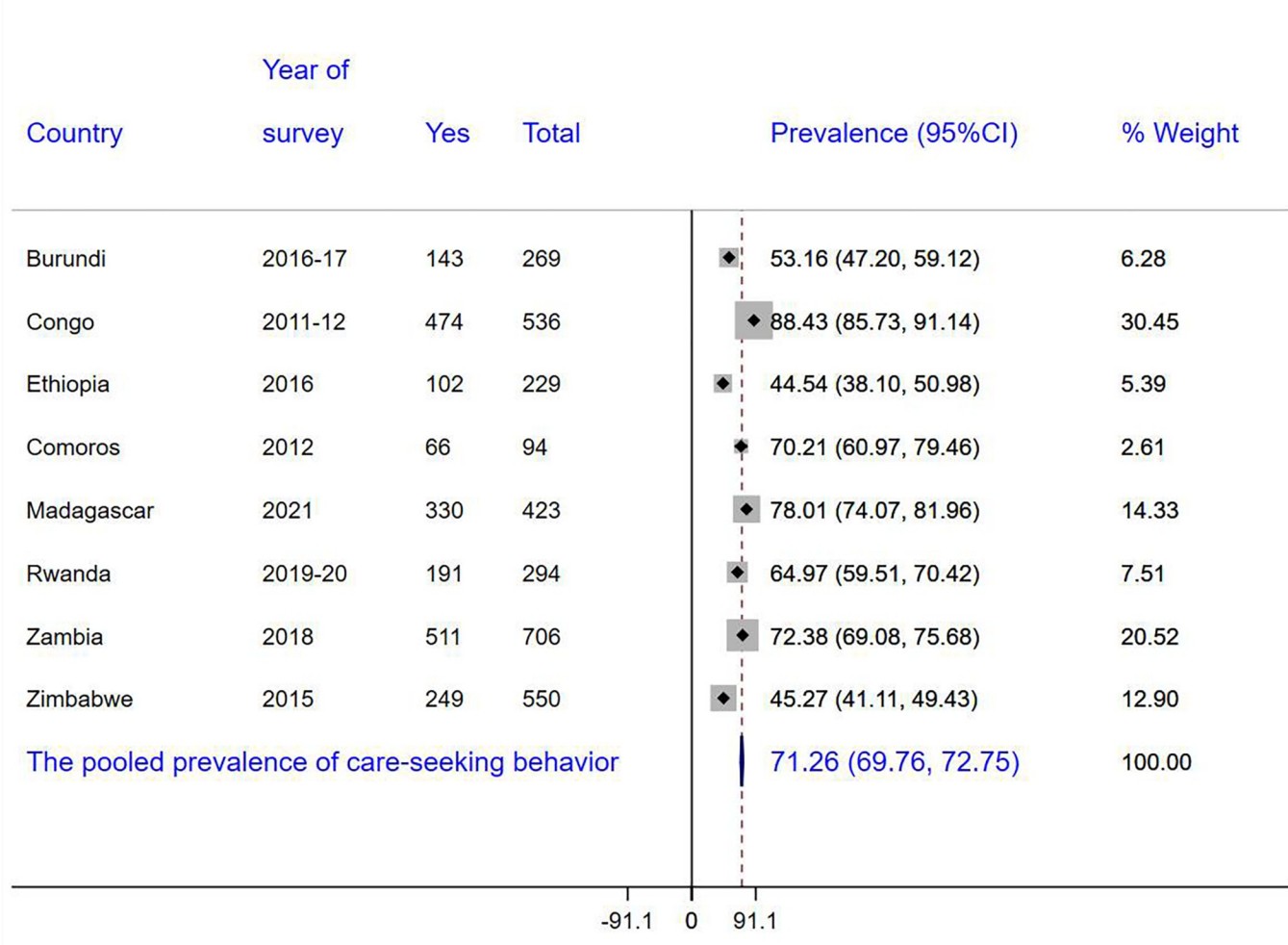

**Fig 1. The pooled prevalence of STI-related care-seeking behavior among sexually active men in East Africa.**

**Table 2. Factors associated with STI-related health-seeking behavior among sexually active men in East Africa.**

| Variable | Null model | Model 1 AOR (95%CI) | Model 2 AOR (95%CI) | Model 3 AOR (95%CI) | P-value |
|---|---|---|---|---|---|
| **Individual level characteristics** | | | | | |
| **Age** | | | | | |
| 15–24 | | 1 | | 1 | <0.005 |
| 25–34 | | 1.77 (1.39, 2.27) | | 1.58 (1.22, 2.04)* | |
| 35–44 | | 1.63 (1.23, 2.16) | | 1.28 (0.95, 1.73) | |
| >44 | | 1.79 (1.29, 2.49) | | 1.44 (1.01, 2.02)* | |
| **Marital status** | | | | | |
| Never married | | 1 | | 1 | <0.000 |
| Ever married | | 1.56 (1.22, 2.00) | | 1.62 (1.25, 2.11)* | |
| **Educational status** | | | | | |
| No formal education | | 1 | | 1 | 0.751 |
| Primary | | 1.13 (0.85, 1.50) | | 0.97 (0.71, 1.32) | |
| Secondary | | 1.41 (1.04, 1.89) | | 1.13 (0.80, 1.60) | |
| Higher | | 0.95 (0.61, 1.48) | | 0.76 (0.47, 1.24) | |
| **Wealth index** | | | | | |
| Poor | | 1 | | 1 | 0.850 |
| Middle | | 0.82 (0.65, 1.03) | | 0.95 (0.75, 1.21) | |
| Rich | | 0.73 (0.59, 0.90) | | 0.98 (0.76, 1.27) | |
| **Employment status** | | | | | |
| Not employed | | 1 | | 1 | 0.331 |
| Employed | | 1.09 (0.85, 1.40) | | 0.87 (0.67, 1.14) | |
| **Age at first sex** | | | | | |
| ≤ 19 | | 1 | | 1 | 0.723 |
| ≥ 20 | | 0.72 (0.59, 0.88) | | 1.04 (0.84, 1.28) | |
| **Number of sex partners excluding spouse, in the last 12 months** | | | | | |
| 0 | | 1 | | 1 | <0.001 |
| 1 | | 2.48 (2.01, 3.07) | | 1.88 (1.50, 2.35)* | |
| ≥2 | | 4.00 (3.04, 5.26) | | 2.53 (1.89, 3.39)* | |
| **Ever been tested for HIV** | | | | | |
| No | | 1 | | 1 | <0.000 |
| Yes | | 1.19 (1.01, 1.42) | | 1.86 (1.52, 2.28)* | |
| **Media exposure** | | | | | |
| No | | 1 | | 1 | <0.050 |
| Yes | | 1.23 (0.99, 1.52) | | 1.30 (1.04, 1.62)* | |
| **Covered by health insurance** | | | | | |
| No | | 1 | | 1 | 0.125 |
| Yes | | 1.19 (0.93, 1.52) | | 1.25 (0.92, 1.70) | |
| **Community level characteristics** | | | | | |
| **Residence** | | | | | |
| Urban | | | 1 | 1 | 0.623 |
| Rural | | | 0.92 (0.77, 1.10) | 1.08 (0.85, 1.37) | |
| **Country** | | | | | |
| Zambia | | | 1 | 1 | <0.000 |
| Burundi | | | 0.42 (0.31, 0.57) | 0.56 (0.39, 0.80)* | |
| Congo | | | 2.95 (2.13, 4.08) | 3.55 (2.48, 5.08)* | |
| Ethiopia | | | 0.29 (0.21, 0.41) | 0.41 (0.28, 0.59)* | |
| Comoros | | | 0.84 (0.51, 1.38) | 1.21 (0.71, 2.06) | |

*(Continued)*

**Table 2.** (Continued)

| Variable | Null model | Model 1 AOR (95%CI) | Model 2 AOR (95%CI) | Model 3 AOR (95%CI) | P-value |
|---|---|---|---|---|---|
| Madagascar | | | 1.36 (1.01, 1.83) | 1.88 (1.34, 2.64)* | |
| Rwanda | | | 0.71 (0.53, 0.97) | 0.69 (0.46, 1.04) | |
| Zimbabwe | | | 0.29 (0.23, 0.38) | 0.28 (0.22, 0.37)* | |
| Random effect analysis and model comparison | | | | | |
| LR test | X$^2$ = 29.11, p-value <0.001 | | | | |
| ICC % | 8.78 | 7.67 | 6.72 | 6.29 | |
| Log-likelihood | -1960.196 | -1853.64 | -1792.651 | -1721.061 | |
| Deviance | 3,920.392 | 3,707.28 | 3,585.302 | 3,442.122 | |
| AIC | 3924.392 | 3743.281 | 3605.302 | 3494.122 | |
| BIC | 3936.471 | 3851.985 | 3665.697 | 3651.14 | |

*p-value<0.05

95%CI: 2.48, 5.08) and Madagascar (AOR = 1.88, 95%CI: 1.34, 2.64) compared to men who reside in Zambia (Table 2).

## Discussion

The pooled prevalence of STI-related care-seeking behavior among sexually active men in East Africa was 71% (95%CI: 69.76, 72.75). The prevalence was higher compared to health care-seeking behavior for STI among women in East Africa 54.44% (95%CI:53.2%,55.0%) (28) and Chinese men who has sex with men (MSM) (51%) [28]. One plausible explanation for the difference in STI care-seeking behavior between genders in East Africa is attributed to the social stigma associated with STIs. This social stigma, prevalent in the region, may act as a deterrent for women to seek medical attention for STIs. The societal norms and cultural practices in East Africa further reinforce the stigma associated with STIs, leading to a higher prevalence of untreated STIs among women in comparison to men. STIs are often stigmatized due to misconceptions and moral judgments. Women may fear being labeled as promiscuous or immoral if they seek STI testing or treatment [26, 29–31].

The fear of social exclusion or rejection prevents many women from seeking help promptly. In some East African communities, people rely on traditional healers, herbal remedies, or spiritual practices for health issues. Women may prefer these methods over seeking professional medical help. Unfortunately, these alternative approaches may not effectively treat STIs, leading to complications. Some cultural beliefs emphasize female virginity and purity. Women who fear judgment or rejection due to STIs may avoid seeking help [32, 33].

When women experience health issues related to sexual intercourse, it is important that they receive prompt care. However, due to their mostly monogamous relationships, they may not immediately associate their complaints with sexual activity. This can hamper their seeking medical attention and potentially worsen their condition [34]. The other possible explanation for higher treatment-seeking behavior among men compared to women may have knowledge regarding STIs, a study done in Ethiopia [35] reported that males were found to have increased levels of knowledge about STIs, which in turn increased their odds of seeking treatment. The disparity with Chinese males could be attributed to a difference in the study population, as the study was conducted on men who had intercourse with men. Although many countries have more progressive policies, there are still some areas across the Middle East that enforce laws against same-sex activity. Unfortunately, these laws prevent MSM (men who have sex with men)

from accessing preventive and curative care in those regions. Even in areas where same-sex activity is legal, the fear of stigma and discrimination can lead men to hide their sexual identity from healthcare professionals. This creates a barrier to testing and preventive treatment [36]. According to the findings of the current study, older men found to have higher odds of treatment-seeking behavior. The possible explanation could be older males may be more aware of the value of preventative treatment and early diagnosis of health issues. Furthermore, older men may have more free time to attend medical appointments and are more likely to have established relationships with healthcare professionals [37]. Consistent with our study, findings from a study done among youths in low and middle-income countries reported that many youths still do not have the information they desire about sexual health or STIs and do not seek timely, formal medical care [38]. One possible explanation for the limited resources available to young people seeking help or information regarding sexual health is the persistence of taboos surrounding sexuality. These taboos make it difficult for young people to access the knowledge and resources they need to seek medical care. As a result, many youths lack knowledge about sexually transmitted infections (STIs) and face challenges in obtaining information and medical assistance [39].

Married men were at higher odds of seeking medical care for their STI symptoms. Previously done study, support this. One argument is that wives wield a greater influence than girlfriends wield and may persuade their husbands to obtain regular checks. Another theory is that married men feel required to keep healthy to provide for their families; simply having a spouse creates a greater sense of duty [40]. Furthermore, we have computed Tukey's adjustment (Tukey's Honestly Significant Difference (HSD) test) to compute p-values and confidence intervals for the pairwise differences. The comparisons are made over the variable marital status with wealth index and health insurance. Married men were found to have more financial means and are likelier to have health insurance than unmarried men are.

Compared to men who do not have multiple sexual partners, men who had multiple sexual partners were found to have increased odds of health care-seeking behavior for STI symptoms. There exist various plausible explanations for this phenomenon. One potential factor could be that men who engage in sexual activity with multiple partners may exhibit a higher degree of awareness concerning the potential health risks associated with sexually transmitted infections (STIs). Consequently, these individuals may be more inclined to seek medical attention when they manifest symptoms suggestive of an STI [41].

Men, who had ever been tested for HIV, like women in East Africa, were more likely to seek medical care for STI symptoms [42]. It is possible that men who have undergone HIV/AIDS testing receive superior counseling and education regarding STIs and their corresponding treatment during their consultation. Such a finding points to the need for the healthcare sector to enhance its counseling and awareness efforts during HIV/AIDS testing, thereby increasing the likelihood that men will seek care for STIs [43].

Compared to sexually active men without media exposure, men who had media exposure had increased odds of medical care-seeking behavior. The utilization of mass media, specifically television and radio, is a vital method for disseminating knowledge and raising awareness of sexually transmitted infections (STIs) to a wide audience. It has been observed that exposure to mass media has a positive correlation with greater knowledge and understanding of STIs. As such, this knowledge enhancement is associated with prompting improved care-seeking behavior for symptoms of STIs [35].

## Strengths and limitations of the study

The study's use of weighted data ensures that it represents the national level, which is a significant strength. Furthermore, the study utilized the most recent DHS data available for each East

African country, leading to current findings. However, this study has some limitations. The DHS data do not give any information regarding the number of individuals approached for interviewee and their response rate cannot be estimated. As it was cross-sectional in nature, it was unable to establish causal effects between the dependent and independent variables. Additionally, respondents had to rely on their recall, creating a possibility for recall bias. During data collection, there could also be social desirability bias while gathering information on healthcare-seeking behavior for STIs from the respondents, which could have influenced how some men reported their outcomes. This bias could have led to an underestimation or overestimation of the odds reported in the study. The analysis results may have been affected due to the disparate data collection periods. In particular, there is a notable difference between the DHS survey data for Rwanda collected in 2019/2020 and Madagascar's data collected in 2021. It is worth mentioning that the COVID-19 pandemic may have influenced both data sets. As a result, the findings should be interpreted with caution, and the potential impact of the pandemic on the data should be taken into account.

## Conclusion

In the current study, men's age, marital status, number of sexual partners, whether they have ever been tested for HIV, and media exposure were identified as statistically significant factors associated with healthcare-seeking behavior for STIs. Based on the findings, it is important to implement policies and strategies aimed at enhancing the care-seeking behavior of young and unmarried men concerning STI symptoms.

Enhancing health promotion and education programs, especially during HIV testing and counseling, is crucial for encouraging care-seeking behavior among men regarding STIs. HIV testing and counseling sessions present a unique opportunity to provide education to individuals about other sexually transmitted infections (STIs). By utilizing these sessions, individuals can gain knowledge about the various types of STIs, which can lead to early detection and treatment of these infections. This approach allows for a more comprehensive approach to healthcare, as it enables healthcare professionals to address multiple health concerns simultaneously. Additionally, individuals who undergo testing and counseling sessions are better equipped to make informed decisions regarding their sexual health, leading to improved health outcomes and a reduction in the spread of STIs. Therefore, it is recommended that testing and counseling sessions be utilized as a vital tool in educating individuals about STIs, and promoting overall health and wellness. Utilizing various media platforms such as TV, radio, and social media to disseminate accurate information about STIs and highlight success stories of young men who sought care and benefited from early diagnosis and treatment is recommended.

It should be noted that a holistic approach involving healthcare providers, policymakers, community leaders, and young men themselves is essential for effective implementation. By addressing these factors, we can create an environment where young men feel empowered to seek care promptly and contribute to reducing the burden of STIs in the region.

## Supporting information

**S1 Table. A bi-variable analysis of factors associated with STI-related health-seeking behavior among sexually active men in East Africa.**
(DOCX)

## Acknowledgments

We acknowledge the MEASURE DHS program for permitting the use of the dataset.

## Author Contributions

**Conceptualization:** Beminate Lemma Seifu.

**Data curation:** Beminate Lemma Seifu, Yohannes Mekuria Negussie.

**Formal analysis:** Beminate Lemma Seifu, Bezawit Melak Fente, Zufan Alamrie Asmare, Hiwot Altaye Asebe, Yohannes Mekuria Negussie.

**Investigation:** Zufan Alamrie Asmare, Angwach Abrham Asnake, Meklit Melaku Bezie.

**Methodology:** Beminate Lemma Seifu, Bezawit Melak Fente, Mamaru Melkam, Zufan Alamrie Asmare, Angwach Abrham Asnake, Meklit Melaku Bezie, Yohannes Mekuria Negussie.

**Software:** Beminate Lemma Seifu, Bezawit Melak Fente, Zufan Alamrie Asmare, Angwach Abrham Asnake, Meklit Melaku Bezie, Hiwot Altaye Asebe, Yohannes Mekuria Negussie.

**Supervision:** Beminate Lemma Seifu, Bezawit Melak Fente, Zufan Alamrie Asmare, Angwach Abrham Asnake, Meklit Melaku Bezie, Hiwot Altaye Asebe, Yohannes Mekuria Negussie.

**Validation:** Beminate Lemma Seifu, Bezawit Melak Fente, Mamaru Melkam, Zufan Alamrie Asmare, Angwach Abrham Asnake, Meklit Melaku Bezie, Hiwot Altaye Asebe, Yohannes Mekuria Negussie.

**Visualization:** Beminate Lemma Seifu, Bezawit Melak Fente, Mamaru Melkam, Zufan Alamrie Asmare, Angwach Abrham Asnake, Meklit Melaku Bezie, Hiwot Altaye Asebe, Yohannes Mekuria Negussie.

**Writing – original draft:** Beminate Lemma Seifu, Bezawit Melak Fente, Mamaru Melkam, Zufan Alamrie Asmare, Angwach Abrham Asnake, Meklit Melaku Bezie, Hiwot Altaye Asebe, Yohannes Mekuria Negussie.

**Writing – review & editing:** Beminate Lemma Seifu, Bezawit Melak Fente, Mamaru Melkam, Zufan Alamrie Asmare, Angwach Abrham Asnake, Meklit Melaku Bezie, Hiwot Altaye Asebe, Yohannes Mekuria Negussie.

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
