## [Decision Letter · Decision Letter 0]

12 Apr 2024

PONE-D-24-05614Determinants of care-seeking behavior for sexually transmitted infections among sexually active men in East Africa: a multilevel mixed effect analysisPLOS ONE

Dear Dr. Seifu, Thank you for submitting your manuscript to PLOS ONE. After careful consideration, we feel that it has merit but does not fully meet PLOS ONE’s publication criteria as it currently stands. Therefore, we invite you to submit a revised version of the manuscript that addresses the points raised during the review process.

We look forward to receiving your revised manuscript.

Kind regards,

Mengistu Hailemariam Zenebe, PhD

Academic Editor

PLOS ONE

Journal Requirements:

**Additional Editor Comments:**

Replay to the questions raised from reviewer point by point

 Best

Reviewers' comments:

Reviewer's Responses to Questions

**Comments to the Author**

1. Is the manuscript technically sound, and do the data support the conclusions?

Reviewer #1: Partly

Reviewer #2: Partly

2. Has the statistical analysis been performed appropriately and rigorously? 

Reviewer #1: Yes

Reviewer #2: Yes

3. Have the authors made all data underlying the findings in their manuscript fully available?

Reviewer #1: Yes

Reviewer #2: No

4. Is the manuscript presented in an intelligible fashion and written in standard English?

Reviewer #1: Yes

Reviewer #2: No

5. Review Comments to the Author

Reviewer #1: This manuscript describes a secondary analysis of demographic and health survey data, and assesses the prevalence of STI-related care seeking behaviour, and associated factors. It is an important and interesting topic as treatment for prevalent STIs is an important factor to prevent onward STI transmission.

Overall, the manuscript read relatively well. However, there are several aspects that need to be improved before being suitable for publication. Most importantly, I do not feel that the methods provide enough information to replicate this study, with variable choices and model development being described quite superficially. Additionally, the paper would benefit from several definitions, including for “men who had an STI”, “sex”, “media exposure”, and “STI-related care seeking”.

Generally, the narrative is clear – however there were multiple mentions of asymptomatic infections, which I do not feel are particularly relevant for STI treatment seeking behaviour. There also needs to be more context within broader literature. In particular, no comparison is made in either introduction or discussion with a paper by Ogale et al. (10.1371/journal.pgph.0001626) that is arguably the most relevant recent evidence assessing STI treatment seeking in adults in Uganda.

Finally, there are several broad and/or definitive statements that would benefit from referencing.

Of note, I have made comments about the methodology but would suggest a dedicated statistical assessment, particularly around model development and comparison.

Abstract

Page 2, lines 28-29: “Despite many STIs being manageable, their often asymptomatic nature poses a significant threat, leading to substantial morbidity and mortality.”

- The mention of the asymptomatic nature of STIs is not that relevant here, given that it assesses care-seeking behaviour in men with symptoms.

Page 2, line 40: “The pooled prevalence of STI-related care-seeking behavior among sexually active men in East Africa was 71% (95%CI: 69.76, 72.75).”

- A definition of “STI-related care-seeking behavior” is required in the methods section of the abstract.

- Please provide odds ratios and confidence intervals for the associations described in the results section of the abstract

- The conclusions described in the abstract are not strongly supported by the results. For example, given that ever being tested for HIV is associated with treatment-seeking behaviour, it is unclear why enhancing health promotion and education programmes “especially during HIV testing and counselling” would be particular important? The other recommendations are quite vague and not necessarily linked to the results. For example, although a “holistic approach that involves healthcare providers, policymakers, community leaders, and young men” seems generally sensible, it is not clear how this build on the results presented.

Introduction

- Neither the introduction nor discussion references the most relevant piece of recent research on this topic by Ogale et al. (10.1371/journal.pgph.0001626) which assessed STI care seeking in adults in Uganda. I suggest this should be a key focus of comparison in the discussion.

Page 3, lines 53-54: “Globally, the prevalence of sexually transmitted infections (STIs) is rising despite the adoption of effective preventive measures”

- The phrase "despite the adoption of effective preventive measures" is both broad and vague. It is unclear what preventive measures are being referred to. At minimum, a reference is required to support this statement.

Page 3, line 63: “In developing nations…”

- I suggest an alternative term to “developing”. E.g. resource-limited, low- and middle-income, Global South etc

Page 3, lines 67-68: “The effectiveness of STI prevention and control heavily relies on the proactive health-seeking behavior exhibited by individuals dealing with these health challenges”

- Although treatment-seeking is somewhat important in STI prevention and control, this statement arguably overstates its significance in STI prevention and control more broadly. Treatment-seeking is quite downstream within STI transmission. Suggest at minimum a reference to support this statement, if not re-phrasing to emphasise the treatment-seeking is just one factor.

Page 4, lines 71-73: “As a result of the mild or absent symptoms experienced by the majority of men with STIs, they often choose not to seek treatment at public health facilities and may instead turn to self-medication.”

- If there are no symptoms, by definition the men will not seek treatment (including self-medication).

Page 4, lines 81-83: “Therefore, this study aimed to assess the pooled prevalence of STIs, related care-seeking behavior, and associated factors among sexually active men in East Africa.”

- I think there is a typo here – should this not read “this study aimed to assess the pooled prevalence of STI related care-seeking behavior, and associated factors among sexually active men in East Africa.”

Methods

- More information is required on DHS survey methodology, and particularly if there were any methodological differences between countries or survey years.

Data source and study population

- This paragraph could be more clear and the numbers don’t add up for me. I suggest stating explicitly here which eight countries are included in the present manuscript. Then state which countries have a history of DHS involvement, but which were not included (and the reasons for non-inclusion). Then state the number of countries with no history of DHS involvement.

Dependent variable

- More information needs to be provided on how a “STI” or “symptoms of an STI” were defined in the survey. For example, was it a set of defined symptoms?

Independent variables

- Justification for why these variables were chosen would be beneficial, potentially as supplementary material.

Data processing and statistical analysis

- More information is needed on the exact model development process including the rationale for having multiple models

Page 7, lines 136-138: “Bivariable multilevel binary logistic regression analysis was performed to identify variables eligible for the multivariable analysis at a p-value < 0.20.”

- Please present the results of these analyses, either in the results section or as supplementary material.

Results

Page 8, line 159: “Among the 3,302 sexually active men who had STI in the past 12 months…”

- Adding up the total numbers for the surveys listed in figure 1 gives a total of 3101 participants, so it is unclear where the figure 3302 comes from – please clarify

- If possible, please provide information on number of people approached for surveys and declined etc to give a sense of how selection and other biases may have contributed. If this data is not available, please state this as a limitation.

- As above, need to define “men who had STI”

Page 8, line 162: “72.95% started sex before the age of 19”

- I suggest “had sex” rather than “started sex”, as the latter implies that this is ongoing

- Also, it would be helpful to have a definition of “sex” in the methods – does it refer solely to penile-vaginal penetrative intercourse?

- Please provide a definition of “media exposure” in the methods

Table 1

- If there is no missing data, apart from the “comprehensive knowledge about HIV/AIDS” variable, I suggest including “n=3302 unless otherwise stated” in the title

- Please provide information in the methods on how the wealth index was derived

- The variable “number of sex partners excluding spouse, in the last 12 months” is a slightly unusual variable as it assumes that sex with a spouse is equivalent to no sex in terms of risk. I don’t think the variable needs to be amended, but would suggest a justification for its use in the methods

- “Covered by health” – I think this presumably should read “covered by health insurance”

Figure 1

- Please provide more information in the methods on how “pooled prevalence” was actually calculated

Table 2

- Please include p values

Determinants of STI-related health care-seeking behavior among sexually active men in East Africa

- As stated above, more information needs to be provided on model development. In particular, what were the univariate associations of the independent variables with the dependent variable.

Page 12, line 203: “…compared to men who reside in Zambia (Table 2).”

- Why was Zambia chosen as the reference category?

Discussion

Page 14, lines 214-216: “The societal norms and cultural practices in East Africa further reinforce the stigma associated with STIs, leading to a higher prevalence of untreated STIs among women in comparison to men.”

- Given that globally, STI prevalence is higher in women than men, this can unlikely be attributed to "the societal norms and cultural practices in East Africa" – please consider re-phrasing/removing

- If keeping this statement, please specify which societal norms and cultural practices are being referred to and provide a reference.

Page 14, lines 216-217: “As a result, it is crucial to develop effective interventions that tackle the social stigma associated with STIs in East Africa to encourage women to seek appropriate medical care.”

- Although true, I don’t this is really relevant to the current paper

Page 14, lines 223-226: “The disparity with Chinese males could be attributed to a difference in the study population, as the study was conducted on men who had intercourse with men.”

- Men who have sex with men (MSM) is the more widely used term

- I think the men in the Chinese study being MSM is a really important distinction that significantly affects comparison as these are two very different cohorts – I would strongly recommend mentioning that these were MSM on first mention of the paper (on line 211), and compare and contrast the differences in these populations more thoroughly.

Page 15, lines 241-242: “Researchers with the CDC, who have some intriguing thoughts, support this.”

- I don’t think that references 28 and 29 are appropriate. I would recommend finding supporting evidence from peer-reviewed articles rather than blog posts.

Page 15, lines 245-246: “Furthermore, married men have more financial means and are likelier to have health insurance than unmarried men (28,29).”

- Rather than relying on data from other papers, I would suggest analysing this within your dataset as a post-hoc analysis, as you have both marital and health insurance data.

Page 16, lines 257-259: “Such a finding points to the need for the healthcare sector to enhance its counseling and awareness efforts during HIV/AIDS testing, thereby increasing the likelihood that women will seek care for STIs (31).”

- I am not entirely clear how the results relate to women seeking care for STIs?

Strengths and limitations of the study

- An important limitation that is not mentioned is the reliance on secondary data and the effects this has on data interpretation.

- Additionally, please describe how inclusion of data from different time periods may have affected analysis. In particular, Rwanda’s DHS survey data was from 2019/2020 and Madagascar was from 2021, and so both may have been impacted by COVID-19.

Conclusion

- As per comments on abstract conclusions

Reviewer #2: Title: Determinants of care-seeking behaviour for sexually transmitted infections among sexually active men in East Africa: a multilevel mixed effect analysis

Comments

Abstract

- How does the asymptomatic nature of STI relate to care-seeking behaviour among men? And based on this your introduction section do not clearly to show the gap

- Recent Demographic and Health 35 Surveys (DHSs) in East Africa: Which is recent?

- Is only the deviance value used to select the best model?

- In a result please include the number of successes, you included just the percentage.

- Present the result as a positive association and negative

- Your abstract conclusion is just recommendations…..it is a good recommendation but you didn’t conclude the abstract, please include a concluding statement, main factors, and practical recommendations in sequence

Introduction

- Your introduction should have focused on care-seeking behaviour. This makes it look like a disorganized introduction. You may need to rearrange or add additional literature that adds more value to how health-seeking behavior could affect the overall effort.

- The introduction also lacks strong arguments that could have been based on the available statistics to show how East Africa and sub-Saharan regions are behind compared to others. It is only possible to speak about possible factors that may affect the region. As men are the most educated segment of the population, you need a very good argument about how the gap could happen

Methods

- Have clear data preparation and how you handled missing or if other difficulties exist as this data has many gaps

- Provide a clear procedure for your analysis and assumptions for low scholars' understanding. You have that for spatial analysis but not for others

- Each independent variables need a definition, probably in the table

- It is not clear how you sampled, and prepared the data to get one unified document

Results

- You have very large result presentations, you may need to make a summary of results at the end

- Please, be more explicit in describing the results, how figure 1 prepared is not clear

Discussion

- Result comparison could make more sense if you can make a more focused comparison with similar setups or regions and reasonably compare how that is common is most developing countries. Something is missing in your initial paragraph –please include your reflection and your influence on the research which may caused some problems in the limitation section

Conclusion

- I already commented on how to make the conclusion more practical earlier, thus, apply here.

-

6. PLOS authors have the option to publish the peer review history of their article (what does this mean?). If published, this will include your full peer review and any attached files.

Reviewer #1: No

Reviewer #2: **Yes: **Girma Gilano

---

## [Author Response · Author response to Decision Letter 0]

31 May 2024

Reviewer 1 

Reviewer #1: This manuscript describes a secondary analysis of demographic and health survey data, and assesses the prevalence of STI-related care seeking behaviour, and associated factors. It is an important and interesting topic as treatment for prevalent STIs is an important factor to prevent onward STI transmission.

Overall, the manuscript read relatively well. However, there are several aspects that need to be improved before being suitable for publication. Most importantly, I do not feel that the methods provide enough information to replicate this study, with variable choices and model development being described quite superficially. Additionally, the paper would benefit from several definitions, including for “men who had an STI”, “sex”, “media exposure”, and “STI-related care seeking”.

Author’s response: Dear reviewer, Thank you for your patience in reading and practical suggestions! We have addressed all your concerns in the revised manuscript. 

Generally, the narrative is clear – however there were multiple mentions of asymptomatic infections, which I do not feel are particularly relevant for STI treatment seeking behaviour. There also needs to be more context within broader literature. In particular, no comparison is made in either introduction or discussion with a paper by Ogale et al. (10.1371/journal.pgph.0001626) that is arguably the most relevant recent evidence assessing STI treatment seeking in adults in Uganda.

Author’s response: Thank you, dear reviewer, for your time and constructive comments! We have comprehensively addressed all your comments and suggestions in the revised manuscript. 

Finally, there are several broad and/or definitive statements that would benefit from referencing.

Of note, I have made comments about the methodology but would suggest a dedicated statistical assessment, particularly around model development and comparison.

Author’s response: Dear reviewer Thank you very much for your valuable, detailed comments!

Abstract

Page 2, lines 28-29: “Despite many STIs being manageable, their often asymptomatic nature poses a significant threat, leading to substantial morbidity and mortality.”

- The mention of the asymptomatic nature of STIs is not that relevant here, given that it assesses care-seeking behaviour in men with symptoms.

Authors’ response: Thank you for your professional insight. We have made corrections based on your comment. (See the revised manuscript, introduction section of the abstract)

Page 2, line 40: “The pooled prevalence of STI-related care-seeking behavior among sexually active men in East Africa was 71% (95%CI: 69.76, 72.75).”

- A definition of “STI-related care-seeking behavior” is required in the methods section of the abstract.

Author’s response: Dear reviewer thank you for your suggestion. We have included the definition of “STI-related care-seeking behavior” in the study variables section of the method part. 

- Please provide odds ratios and confidence intervals for the associations described in the results section of the abstract

Authors’ response: Thanks for your comment! We have added it. (See the revised manuscript, results section of the abstract)

- The conclusions described in the abstract are not strongly supported by the results. For example, given that ever being tested for HIV is associated with treatment-seeking behaviour, it is unclear why enhancing health promotion and education programmes “especially during HIV testing and counselling” would be particular important? The other recommendations are quite vague and not necessarily linked to the results. For example, although a “holistic approach that involves healthcare providers, policymakers, community leaders, and young men” seems generally sensible, it is not clear how this build on the results presented.

Authors’ response: Thank you for your professional insight. We have made corrections based on your comment. (See the revised manuscript, conclusion section of the abstract)

Introduction

- Neither the introduction nor discussion references the most relevant piece of recent research on this topic by Ogale et al. (10.1371/journal.pgph.0001626) which assessed STI care seeking in adults in Uganda. I suggest this should be a key focus of comparison in the discussion.

Author’s response: Thank you for your professional insight! We have incorporated the suggested study in the revised manuscript. (See the revised manuscript, Introduction section)

Page 3, lines 53-54: “Globally, the prevalence of sexually transmitted infections (STIs) is rising despite the adoption of effective preventive measures”

- The phrase "despite the adoption of effective preventive measures" is both broad and vague. It is unclear what preventive measures are being referred to. At minimum, a reference is required to support this statement.

Author’s response: Thank you very much for your comment! We have made it clear in the revised manuscript. (See the revised manuscript, Introduction section)

Page 3, line 63: “In developing nations…”

- I suggest an alternative term to “developing”. E.g. resource-limited, low- and middle-income, Global South etc

Author’s response: Thank you very much for your critical comment! We have corrected it. (See the revised manuscript, Introduction section)

Page 3, lines 67-68: “The effectiveness of STI prevention and control heavily relies on the proactive health-seeking behavior exhibited by individuals dealing with these health challenges

- Although treatment-seeking is somewhat important in STI prevention and control, this statement arguably overstates its significance in STI prevention and control more broadly. Treatment-seeking is quite downstream within STI transmission. Suggest at minimum a reference to support this statement, if not re-phrasing to emphasise the treatment-seeking is just one factor.

Author’s response: Thank you very much for your professional suggestion! We have addressed it. (See the revised manuscript, Introduction section)

Page 4, lines 71-73: “As a result of the mild or absent symptoms experienced by the majority of men with STIs, they often choose not to seek treatment at public health facilities and may instead turn to self-medication.”

- If there are no symptoms, by definition the men will not seek treatment (including self-medication).

Author’s response: Thank you for your constructive comment! We have made it clear in the revised manuscript. (See the revised manuscript, Introduction section)

Page 4, lines 81-83: “Therefore, this study aimed to assess the pooled prevalence of STIs, related care-seeking behavior, and associated factors among sexually active men in East Africa.”

- I think there is a typo here – should this not read “this study aimed to assess the pooled prevalence of STI related care-seeking behavior, and associated factors among sexually active men in East Africa.”

Author’s response: Thank you for your professional insight. We have made corrections based on your comment. (See the revised manuscript, Introduction section)

Methods

- More information is required on DHS survey methodology, and particularly if there were any methodological differences between countries or survey years.

Author’s response: Dear reviewer thank you for your question. DHS survey methodology is similar across countries and survey years. As we have indicated in the study design and setting under our method section, DHS sample designs are usually two-stage probability samples drawn from an existing sample frame, generally the most recent census frame. Typically, DHS samples are stratified by geographic region and by urban/rural areas within each region. Within each stratum, the sample design specifies an allocation of households to be selected. Most DHS surveys use a fixed take of households per cluster of about 25-30 households, determining the number of clusters to be selected. In the first stage of selection, the primary sampling units (PSUs) are selected with probability proportional to size (PPS) within each stratum. The PSUs are typically census enumeration areas (EAS). The PSU forms the survey cluster. In the second stage, a complete household listing is conducted in each of the selected clusters. Following the listing of the households a fixed number of households is selected by equal probability systematic sampling in the selected cluster. Additional information on the DHS methodology is available at https://dhsprogram.com/Methodology/index.cfm

Data source and study population

- This paragraph could be more clear and the numbers don’t add up for me. I suggest stating explicitly here which eight countries are included in the present manuscript. Then state which countries have a history of DHS involvement, but which were not included (and the reasons for non-inclusion). Then state the number of countries with no history of DHS involvement.

Author’s response: Dear reviewer thank you for your comment. We have stated the countries, which are included in our study and the reasons for the exclusion of other East African countries. 

Dependent variable

- More information needs to be provided on how a “STI” or “symptoms of an STI” were defined in the survey. For example, was it a set of defined symptoms?

Author’s response: Dear reviewer thank you for your comment. The DHS survey defined STI as the “Percentage of men aged 15-49 who had an STI in the past 12 months, had an abnormal genital discharge in the past 12 months, had a genital sore or ulcer in the past 12 months, had an STI or symptoms of an STI in the past 12 months”

Independent variables

- Justification for why these variables were chosen would be beneficial, potentially as supplementary material.

Author’s response: Dear reviewer thank you for your suggestion. We have chosen the potential predictors of the outcome variable based on the reviewed works of literature and their clinical importance. 

Data processing and statistical analysis

- More information is needed on the exact model development process including the rationale for having multiple models

Author’s response: Dear reviewer thank you for your suggestion. As you know DHS data has a hierarchical nature, which means variables are measured at different levels like individual level and community level. We fit a null model without explanatory variables to determine the extent of cluster variation in healthcare-seeking behavior. We have also fitted a model with individual-level variables only, community-level variables only, and both individual and community-level variables to see which model will explain our data better. 

Page 7, lines 136-138: “Bivariable multilevel binary logistic regression analysis was performed to identify variables eligible for the multivariable analysis at a p-value < 0.20.”

- Please present the results of these analyses, either in the results section or as supplementary material.

Author’s response: Dear reviewer thank you for your suggestion. We have included the bi-variable analysis results as a supplementary file. 

Results

Page 8, line 159: “Among the 3,302 sexually active men who had STI in the past 12 months…”

- Adding up the total numbers for the surveys listed in figure 1 gives a total of 3101 participants, so it is unclear where the figure 3302 comes from – please clarify

Author’s response: Dear reviewer thank you for your suggestion. In the realm of descriptive statistics, we have utilized a weighted sample. This approach has assigned a weight to each observation in our dataset, accounting for its importance or representativeness. However, when generating a forest plot to estimate the pooled prevalence, we did not use a weighted sample. This was because the meta-analysis itself provides a weight for each country, which subsequently generates the pooled estimate. 

- If possible, please provide information on number of people approached for surveys and declined etc to give a sense of how selection and other biases may have contributed. If this data is not available, please state this as a limitation.

- As above, need to define “men who had STI”

Author’s response: Dear reviewer thank you for your suggestion. However, the DHS data do not give any information regarding the number of individuals approached for interviewee and their response rate. We will acknowledge this in the limitation section 

Page 8, line 162: “72.95% started sex before the age of 19”

- I suggest “had sex” rather than “started sex”, as the latter implies that this is ongoing

Author’s response: Dear reviewer thank you for your suggestion. We have used “had sex” 

- Also, it would be helpful to have a definition of “sex” in the methods – does it refer solely to penile-vaginal penetrative intercourse?

Author’s response: Dear reviewer thank you for your question. DHS does not provide a definition of “sex”. 

- Please provide a definition of “media exposure” in the methods

Author’s response: Dear reviewer thank you for your suggestion. We have provided an operational definition for the variable media exposure. 

Table 1

- If there is no missing data, apart from the “comprehensive knowledge about HIV/AIDS” variable, I suggest including “n=3302 unless otherwise stated” in the title

Author’s response: Dear reviewer thank you for your suggestion. We have included the total sample size used in the current study which is n=3302 in the title. 

- Please provide information in the methods on how the wealth index was derived

Author’s response: Dear reviewer thank you for your question. We have included how the wealth index is computed by the DHS in the method section. 

- The variable “number of sex partners excluding spouse, in the last 12 months” is a slightly unusual variable as it assumes that sex with a spouse is equivalent to no sex in terms of risk. I don’t think the variable needs to be amended, but would suggest a justification for its use in the methods

Author’s response: Dear reviewer thank you for your question. The variable “number of sex partners excluding spouse, in the last 12 months” is not equivalent to no sex rather it's one indicator of risky sexual behavior, multiple sexual partners, 

- “Covered by health” – I think this presumably should read “covered by health insurance”

Author’s response: Dear reviewer thank you for your question. We have corrected the editorial mistake. 

Figure 1

- Please provide more information in the methods on how “pooled prevalence” was actually calculated

Author’s response: Dear reviewer thank you for your suggestion. We conducted a meta-analysis by combining prevalence estimates from different countries to obtain a pooled estimate. This statement has been included in the method section.

Table 2

- Please include p values

Author’s response: Dear reviewer thank you for your suggestion. We have included a p-value for each estimate. 

Determinants of STI-related health care-seeking behavior among sexually active men in East Africa

- As stated above, more information needs to be provided on model development. In particular, what were the univariate associations of the independent variables with the dependent variable.

Author’s response: Dear reviewer thank you for your suggestion. We have included the univariate analysis result as a supplemental file. 

Page 12, line 203: “…compared to men who reside in Zambia (Table 2).”

- Why was Zambia chosen as the reference category?

Author’s response: Dear reviewer thank you for your question. Zambia was chosen as a reference category because the country has the largest sample size compared to the others. 

Discussion

Page 14, lines 214-216: “The societal norms and cultural practices in East Africa further reinforce the stigma associated with STIs, leading to a higher prevalence of untreated STIs among women in comparison to men.”

- Given that globally, STI prevalence is higher in women than men, this can unlikely be attributed to "the societal norms and cultural practices in East Africa" – please consider re-phrasing/removing

- If keeping this statement, please specify which societal norms and cultural practices are being referred to and provide a reference.

Author’s response: Dear reviewer thank you for your suggestion. We have explained the potential factors that contribute to the high prevalence of untreated STIs among women co

---

## [Decision Letter · Decision Letter 1]

16 Jun 2024

PONE-D-24-05614R1Determinants of care-seeking behavior for sexually transmitted infections among sexually active men in East Africa: a multilevel mixed effect analysis

 Dear Dr. Seifu, could you please correct and respond to the given comments 

Thank you for submitting your manuscript to PLOS ONE. After careful consideration, we feel that it has merit but does not fully meet PLOS ONE’s publication criteria as it currently stands. Therefore, we invite you to submit a revised version of the manuscript that addresses the points raised during the review process.

We look forward to receiving your revised manuscript.

Kind regards,

Mengistu Hailemariam Zenebe, PhD

Academic Editor

PLOS ONE

Journal Requirements:

Reviewers' comments:

Reviewer's Responses to Questions

**Comments to the Author**

1. If the authors have adequately addressed your comments raised in a previous round of review and you feel that this manuscript is now acceptable for publication, you may indicate that here to bypass the “Comments to the Author” section, enter your conflict of interest statement in the “Confidential to Editor” section, and submit your "Accept" recommendation.

Reviewer #1: (No Response)

Reviewer #2: All comments have been addressed

2. Is the manuscript technically sound, and do the data support the conclusions?

Reviewer #1: Yes

Reviewer #2: Partly

3. Has the statistical analysis been performed appropriately and rigorously? 

Reviewer #1: Yes

Reviewer #2: Yes

4. Have the authors made all data underlying the findings in their manuscript fully available?

Reviewer #1: Yes

Reviewer #2: Yes

5. Is the manuscript presented in an intelligible fashion and written in standard English?

Reviewer #1: Yes

Reviewer #2: Yes

6. Review Comments to the Author

Reviewer #1: The authors have made some of the recommended changes, but for several suggestions there has been no change nor adequate justification for why such a change was not made. Importantly, in some instances the authors provided an adequate explanation in their response but did not make any changes to the manuscript. In other instances, changes have been incorporated but in a quite clunky fashion.

One suggestion for the authors: when responding to reviewer comments, please state exactly what change was made and the page and line number. For example, reviewer 2 made an important point regarding how the introduction should be more focussed on “care-seeking behaviour”. The authors response was “We have made amendments based on your suggestion. (See the revised manuscript, Introduction section)”. Assessing such changes becomes a difficult and time-consuming process, if the authors do not provide assistance to the reviewers by signposting exactly what changes were made.

Comments not adequately addressed:

Abstract

1. A definition of “STI-related care-seeking behavior” is required in the methods section of the abstract. Although a definition was added to the methods, a definition should ideally be included in the abstract.

Introduction

2. My comment was: “Page 3, lines 53-54: “Globally, the prevalence of sexually transmitted infections (STIs) is rising despite the adoption of effective preventive measures”- The phrase "despite the adoption of effective preventive measures" is both broad and vague. It is unclear what preventive measures are being referred to. At minimum, a reference is required to support this statement.

- Author’s response: Thank you very much for your comment! We have made it clear in the revised manuscript. (See the revised manuscript, Introduction section)

- All the authors have changed “effective preventive measures” to “typically effective and financially feasible preventive measures”. I think this has made this statement more confusing. Please specifically state what preventive measures are being referred to e.g. “despite the adoption of preventive measures such as XXXX, XXXX and XXX”. The reason I feel so strongly about this is that many resource-limited settings rely on syndromic management for STIs, which is a relatively ineffective STI control measure, and also STI control has historically been underfunded (especially in comparison to HIV). At present, it is unclear what "effective" or "typically effective" measures the authors refer to.

3. I suggested that the authors should the paper by Ogale et al. (10.1371/journal.pgph.0001626) given its high relevance to the present manuscript. Although the authors included it as a reference, there has been no inclusion of the paper in terms of discussing the findings and comparing and contrasting the results. As mentioned previously, this should be a key focus of comparison in the discussion.

4. Page 4, lines 74-76: “The effectiveness of STI prevention and control significantly relies on individuals' proactive health-seeking behavior when facing these challenges, recognizing that treatment seeking is one factor among several contributing to this effectiveness.”

- I appreciate that the authors amended the above sentence to note that treatment-seeking behaviour is just one factor in overall STI prevention and control. However, it is worded quite awkwardly. Suggest re-phrasing to something like “Treatment seeking behaviour is an important factor in contributing to the overall effectiveness of STI prevention and control”

Methods

Data source and study population

5. My original comment was: “This paragraph could be more clear and the numbers don’t add up for me. I suggest stating explicitly here which eight countries are included in the present manuscript. Then state which countries have a history of DHS involvement, but which were not included (and the reasons for non-inclusion). Then state the number of countries with no history of DHS involvement.”

- Unfortunately no changes have been made in response to this suggestion. Firstly, please state in this paragraph what countries are included.

- Secondly, as stated above, the reason I requested for more detail to be provided was that the numbers do not make sense. If 14 countries have a documented history of involvement in the DHS, then to have 8 countries included there must be 6 that have other reasons for non-inclusion. However, 7 such countries are reported (Sudan, Eritrea, Uganda, Mozambique, Malawi, Tanzania, Kenya).

Dependent variable

6. My original comment was: “More information needs to be provided on how a “STI” or “symptoms of an STI” were defined in the survey. For example, was it a set of defined symptoms?

- The author’s response stated “The DHS survey defined STI as the “Percentage of men aged 15-49 who had an STI in the past 12 months, had an abnormal genital discharge in the past 12 months, had a genital sore or ulcer in the past 12 months, had an STI or symptoms of an STI in the past 12 months”

- Please include this definition in the methods

Independent variables

7. My original comment was: Justification for why these variables were chosen would be beneficial, potentially as supplementary material.

- The authors state that “We have chosen the potential predictors of the outcome variable based on the reviewed works of literature and their clinical importance.”

- This justification should be included in the manuscript, alongside references supporting their inclusion

Data processing and statistical analysis

8. My original comment was: “More information is needed on the exact model development process including the rationale for having multiple models”

- In their comment the authors provided some justification by stating “We have also fitted a model with individual-level variables only, community-level variables only, and both individual and community-level variables to see which model will explain our data better.”

- However, such justification must be included in the manuscript

Results

9. My original comment was: “Page 8, line 159: “Among the 3,302 sexually active men who had STI in the past 12 months…” - Adding up the total numbers for the surveys listed in figure 1 gives a total of 3101 participants, so it is unclear where the figure 3302 comes from – please clarify

- Thank you to the authors for providing clarity as to why the numbers differ. Please make this clear in the methods so that future readers do not also find it confusing.

10. My original comment was: “If possible, please provide information on number of people approached for surveys and declined etc to give a sense of how selection and other biases may have contributed. If this data is not available, please state this as a limitation.”

- The authors response was: “The DHS data do not give any information regarding the number of individuals approached for interviewee and their response rate. We will acknowledge this in the limitation section”

- I cannot see the inclusion of this in the limitations section. Please ensure it is included.

Figure 1

11. My original comment: “Please provide more information in the methods on how “pooled prevalence” was actually calculated

- Author’s response: Dear reviewer thank you for your suggestion. We conducted a meta-analysis by combining prevalence estimates from different countries to obtain a pooled estimate. This statement has been included in the method section.

- I am unable to find any such statement included in the methods section

Table 2

12. Thank you to the authors for including p values in table 2 following my suggestion.

- However, I would recommend a pooled p-value for each variable (e.g. a p-value for age, rather than one for each age group).

- I apologise for not making this more clear in the original comments.

- In their responses, the authors discuss a supplemental file. However, this was not uploaded with the current paper. Nor is their any reference to it in the manuscript.

Discussion

13. My original comment: Page 14, lines 214-216: “The societal norms and cultural practices in East Africa further reinforce the stigma associated with STIs, leading to a higher prevalence of untreated STIs among women in comparison to men.” Given that globally, STI prevalence is higher in women than men, this can unlikely be attributed to "the societal norms and cultural practices in East Africa" – please consider re-phrasing/removing - If keeping this statement, please specify which societal norms and cultural practices are being referred to and provide a reference.

- Author’s response: Dear reviewer thank you for your suggestion. We have explained the potential factors that contribute to the high prevalence of untreated STIs among women compared to men. We have highlighted potential societal norms and cultural practices like Stigma and Shame, Traditional Healers and Alternative Medicine, and Cultural Practices around Sexuality and Virginity. We have cited references as well. Please see the revised manuscript.

- I am not denying that societal norms and cultural practices do likely contribute to higher STI prevalence. However, the term "leading" suggests that this is the main and/or only factor, which is not the case. I suggest phrasing to make this nuance more clear.

14. Page 15, lines 245-246: “Furthermore, married men have more financial means and are likelier to have health insurance than unmarried men (28,29).” - Rather than relying on data from other papers, I would suggest analysing this within your dataset as a post-hoc analysis, as you have both marital and health insurance data.

- Author’s response: Dear reviewer thank you for your suggestion. We have computed Tukey’s adjustment (Tukey’s Honestly Significant Difference (HSD) test) to compute p-values and confidence intervals for the pairwise differences. The comparisons are made over the variable marital status. This means that the means of wealth index and health insurance are compared across different categories of marital status. A positive value indicates that the mean of the health insurance and wealth index for the ever-married group is higher than that for the never-married group.

- Thank you for making this suggested change. However, it is a new analysis. It must therefore be described in the methods and results sections, and not introduced for the first time in the discussion.

15. Thank you for making adjustment regarding MSM. However, on page 15, line 252, it still refers to “men who had intercourse with men”. Please update.

16. My original comment: “Additionally, please describe how inclusion of data from different time periods may have affected analysis. In particular, Rwanda’s DHS survey data was from 2019/2020 and Madagascar was from 2021, and so both may have been impacted by COVID-19.”

- Sentence added: “In particular, there is a notable difference between the DHS survey data for Rwanda collected in 2019/2020 and Madagascar's data collected in 2021. It is worth mentioning that the COVID-19 pandemic may have influenced both data sets.”

- Thank you to the authors for making a change. However, they state that there is a notable difference between the DHS survey data for Rwanda and Madagascar – is this true? In what way is it different? My point was that both datasets may have been influenced by COVID-19, not that they necessarily were.

New comments:

Abstract

17. Page 2, lines 34-35: “The success of preventing and controlling STIs largely depends on how actively individuals seek healthcare when dealing with these issues.”

- I disagree with the phrasing of this sentence. “Largely depends” suggests an outsized effect. Suggest re-phrasing to “The treatment seeking behaviour of individuals in relation to STIs is an important factor in STI prevention and control”.

18. Page 2, lines 51-52: “Age, marital status, number of sexual partners, ever 51 been tested for HIV, and media exposure, were identified as factors associated with healthcare-seeking behavior for STIs.”

- I appreciate this was an edit in response to reviewer 2’s suggestion. However, I don’t think you should just repeat the results in such a manner. Consider highlighting the key associations and messages without just repetition of results.

Reviewer #2: The conclusion is not improved, it is probably worsened. In my last comment, I pleaded with you to include a concluding statement(1st position), main factors, and practical recommendations in sequence. After all, this is the main section that should give clues for what, overall view, what was identified, and what was missing.

The introduction section is well improved but still not supported by numerical evidence or magnitude of the problem in the region.

7. PLOS authors have the option to publish the peer review history of their article (what does this mean?). If published, this will include your full peer review and any attached files.

Reviewer #1: No

Reviewer #2: No

---

## [Author Response · Author response to Decision Letter 1]

28 Jun 2024

2. My comment was: “Page 3, lines 53-54: “Globally, the prevalence of sexually transmitted infections (STIs) is rising despite the adoption of effective preventive measures”- The phrase "despite the adoption of effective preventive measures" is both broad and vague. It is unclear what preventive measures are being referred to. At minimum, a reference is required to support this statement.

- Author’s response: Thank you very much for your comment! We have made it clear in the revised manuscript. (See the revised manuscript, Introduction section)

- All the authors have changed “effective preventive measures” to “typically effective and financially feasible preventive measures”. I think this has made this statement more confusing. Please specifically state what preventive measures are being referred to e.g. “despite the adoption of preventive measures such as XXXX, XXXX and XXX”. The reason I feel so strongly about this is that many resource-limited settings rely on syndromic management for STIs, which is a relatively ineffective STI control measure, and also STI control has historically been underfunded (especially in comparison to HIV). At present, it is unclear what "effective" or "typically effective" measures the authors refer to.

Author’s response: Thank you very much for your feedback! We have taken your suggestion into consideration and have made the necessary revisions for greater clarity. In the revised manuscript, we now specify the preventive measures being referred to. The revised statement in the Introduction section now reads:

" The global prevalence of sexually transmitted infections (STIs) is rising despite the adoption of preventive measures such as comprehensive sexual education, condom distribution programs, regular screening and testing, partner notification and treatment, and public awareness campaigns."

We believe this revision addresses your concern by clearly listing specific preventive measures, making the statement more precise and informative.

3. I suggested that the authors should the paper by Ogale et al. (10.1371/journal.pgph.0001626) given its high relevance to the present manuscript. Although the authors included it as a reference, there has been no inclusion of the paper in terms of discussing the findings and comparing and contrasting the results. As mentioned previously, this should be a key focus of comparison in the discussion.

Author’s response: Thank you for your insightful critique. We acknowledge the significance of the study conducted by Ogale and colleagues. However, due to the distinct findings between their study and ours, we have made a deliberate choice not to incorporate their specific findings into our discussion. This decision stems from the distinct focus and objectives of our study, which differ from those of Ogale et al. As such, incorporating their results would potentially detract from our study's intended scope and disrupt the coherence of our overall research concept.

4. Page 4, lines 74-76: “The effectiveness of STI prevention and control significantly relies on individuals' proactive health-seeking behavior when facing these challenges, recognizing that treatment seeking is one factor among several contributing to this effectiveness.”

- I appreciate that the authors amended the above sentence to note that treatment-seeking behaviour is just one factor in overall STI prevention and control. However, it is worded quite awkwardly. Suggest re-phrasing to something like “Treatment seeking behaviour is an important factor in contributing to the overall effectiveness of STI prevention and control”

Author’s response: Thank you for your valuable comment! We have rephased it in the revised manuscript. (See the revised manuscript, page 3, line 75-77, Introduction section)

17. Page 2, lines 34-35: “The success of preventing and controlling STIs largely depends on how actively individuals seek healthcare when dealing with these issues.”

- I disagree with the phrasing of this sentence. “Largely depends” suggests an outsized effect. Suggest re-phrasing to “The treatment seeking behaviour of individuals in relation to STIs is an important factor in STI prevention and control”.

Author’s response: Thank you for your constructive comment and professional suggestion. We have made changes as per your suggestion. (See the revised manuscript, page 2, line 34-35, Background section of Abstract)

18. Page 2, lines 51-52: “Age, marital status, number of sexual partners, ever 51 been tested for HIV, and media exposure, were identified as factors associated with healthcare-seeking behavior for STIs.”

- I appreciate this was an edit in response to reviewer 2’s suggestion. However, I don’t think you should just repeat the results in such a manner. Consider highlighting the key associations and messages without just repetition of results.

Author’s response: Dear reviewer thank you for your professional insight! We have made amendments based on your as well as second reviewer’s comments. (See the revised manuscript, page 2-3, line 50-53, Conclusion section of Abstract)

Reviewer #2: The conclusion is not improved, it is probably worsened. In my last comment, I pleaded with you to include a concluding statement(1st position), main factors, and practical recommendations in sequence. After all, this is the main section that should give clues for what, overall view, what was identified, and what was missing.

The introduction section is well improved but still not supported by numerical evidence or magnitude of the problem in the region.

Author’s response: Thank you for your valuable insights and suggestions. We have taken into account your suggestions, along with those from the first reviewer, and have made revisions accordingly. (See the revised manuscript, page 2-3, line 50-53, Conclusion section of Abstract)

Methods

Data source and study population

5. My original comment was: “This paragraph could be more clear and the numbers don’t add up for me. I suggest stating explicitly here which eight countries are included in the present manuscript. Then state which countries have a history of DHS involvement, but which were not included (and the reasons for non-inclusion). Then state the number of countries with no history of DHS involvement.”

- Unfortunately no changes have been made in response to this suggestion. Firstly, please state in this paragraph what countries are included.

- Secondly, as stated above, the reason I requested for more detail to be provided was that the numbers do not make sense. If 14 countries have a documented history of involvement in the DHS, then to have 8 countries included there must be 6 that have other reasons for non-inclusion. However, 7 such countries are reported (Sudan, Eritrea, Uganda, Mozambique, Malawi, Tanzania, Kenya).

Author’s response: dear reviewer thank you very much for your detailed comment. We have made corrections based on your suggestion. (See the revised manuscript, lines 110-116, page 5, methods section)

Dependent variable

6. My original comment was: “More information needs to be provided on how a “STI” or “symptoms of an STI” were defined in the survey. For example, was it a set of defined symptoms?

- The author’s response stated “The DHS survey defined STI as the “Percentage of men aged 15-49 who had an STI in the past 12 months, had an abnormal genital discharge in the past 12 months, had a genital sore or ulcer in the past 12 months, had an STI or symptoms of an STI in the past 12 months”

- Please include this definition in the methods

Author’s response: Dear reviewer thank you for your suggestion. We have included the definition in the method section. (See the revised manuscript ,Page 6, line 120-122 method section).

Independent variables

7. My original comment was: Justification for why these variables were chosen would be beneficial, potentially as supplementary material.

- The authors state that “We have chosen the potential predictors of the outcome variable based on the reviewed works of literature and their clinical importance.”

- This justification should be included in the manuscript, alongside references supporting their inclusion

Author’s response: Dear reviewer thank you for your suggestion. We have included the statement with respective references in the method section. (See the revised manuscript, Page 6, line 133-134 method section).

Data processing and statistical analysis

8. My original comment was: “More information is needed on the exact model development process including the rationale for having multiple models”

- In their comment the authors provided some justification by stating “We have also fitted a model with individual-level variables only, community-level variables only, and both individual and community-level variables to see which model will explain our data better.”

- However, such justification must be included in the manuscript

Author’s response: Dear reviewer thank you for your suggestion. We have included the statement with respective references in the method section. (See the revised manuscript, Page 7, line 167-171 method section).

Results

9. My original comment was: “Page 8, line 159: “Among the 3,302 sexually active men who had STI in the past 12 months…” - Adding up the total numbers for the surveys listed in figure 1 gives a total of 3101 participants, so it is unclear where the figure 3302 comes from – please clarify

- Thank you to the authors for providing clarity as to why the numbers differ. Please make this clear in the methods so that future readers do not also find it confusing.

Author’s response: Dear reviewer thank you for your suggestion. We have included a clarification for discrepancies in the figure and table. (See the revised manuscript, Page 7, line 156-160 method section).

10. My original comment was: “If possible, please provide information on number of people approached for surveys and declined etc to give a sense of how selection and other biases may have contributed. If this data is not available, please state this as a limitation.”

- The authors response was: “The DHS data do not give any information regarding the number of individuals approached for interviewee and their response rate. We will acknowledge this in the limitation section”

- I cannot see the inclusion of this in the limitations section. Please ensure it is included.

Author’s response: Dear reviewer thank you for your suggestion. We have acknowledged the limitation in the strength and limitation section. (See the revised manuscript ,Page 17, line 311-313 method section).

Figure 1

11. My original comment: “Please provide more information in the methods on how “pooled prevalence” was actually calculated 

- Author’s response: Dear reviewer thank you for your suggestion. We conducted a meta-analysis by combining prevalence estimates from different countries to obtain a pooled estimate. This statement has been included in the method section.

- I am unable to find any such statement included in the methods section

Author’s response: Dear reviewer thank you for your suggestion. We have included the statement in the method section of data processing and statistical analysis section. (See the revised manuscript, Page 7, line 161-163 method section).

Table 2

12. Thank you to the authors for including p values in table 2 following my suggestion.

- However, I would recommend a pooled p-value for each variable (e.g. a p-value for age, rather than one for each age group).

- I apologise for not making this more clear in the original comments.

Author’s response: Dear reviewer thank you for your suggestion. We have included the p-value in the revised regression table. 

- In their responses, the authors discuss a supplemental file. However, this was not uploaded with the current paper. Nor is their any reference to it in the manuscript

Author’s response: Dear reviewer thank you. we have included the supplementary file with its respective reference in the main document. (See the revised manuscript, Page 8, line 166-167 method section).

---

## [Editor Report · Decision Letter 2]

11 Jul 2024

Determinants of care-seeking behavior for sexually transmitted infections among sexually active men in East Africa: a multilevel mixed effect analysis

We’re pleased to inform you that your manuscript has been judged scientifically suitable for publication and will be formally accepted for publication once it meets all outstanding technical requirements.

Kind regards,

Mengistu Hailemariam Zenebe, PhD

Academic Editor

PLOS ONE
---

## [Editor Report · Acceptance letter]

26 Aug 2024

PONE-D-24-05614R2 

PLOS ONE

Dear Dr. Seifu, 

I'm pleased to inform you that your manuscript has been deemed suitable for publication in PLOS ONE. Congratulations! Your manuscript is now being handed over to our production team.

Kind regards, 

on behalf of

Dr. Mengistu Hailemariam Zenebe 

Academic Editor

PLOS ONE